# Invented Communities and Social Vulnerability: The Local Post-Disaster Dynamics of Extreme Environmental Events

**Rolf Lidskog** 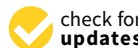

Environmental Sociology Section, School of Humanities, Educational and Social Sciences, Örebro University, 702 81 Örebro, Sweden; rolf.lidskog@oru.se; Tel.: +46-19-303-272

**Abstract:** This paper investigates post-disaster dynamics at the local level, in particular how local identity and social cohesion are affected after an extreme event. A particular case is investigated: the largest forest fire in modern Swedish history, which took place in 2014. The empirical material consists of interviews with forest professionals and organizations involved with the fire or the post-fire work and a postal survey to all people directly affected by the wildfire. The analysis finds that the experience of the wildfire and its social interpretation led to the invention of a particular community identity, one that strengthened the self-understanding of the community. Thus, the post-disaster dynamics are pivotal for what social practices that emerge and what local identities are invented and thus may greatly affect the capacity of a community to handle extreme events.

**Keywords:** community resilience; disasters; extreme events; forest fires; invented communities; social vulnerability

## 1. Introduction

Extreme events are increasingly prominent in society. Media constantly reports about environmental disasters and they are topics of public and political discussion. Because many extreme events are not fully avoidable or are too costly to avoid, much discussion concerns how to minimize their consequences. A current global challenge is therefore to decrease the vulnerability and increase the resilience of socio-ecological systems in order to minimize the adverse effects of extreme events. This is also fuelled by prognoses that extreme weather will probably be more common in the future due to climate change [1]. In this discussion, vulnerability and resilience at the community level have increasingly come into focus, not least because localities on all continents are exposed to extreme environmental events, such as storms and flooding.

A number of studies have focused on how to make communities more resilient and better prepared to face disasters [2–5]. This paper takes a different path. Instead of investigating how a community can become less vulnerable to extreme events, it will study how an extreme event has affected a particular community, not least its local identity. The reason for this emphasis is that earlier research has found that communities can have different responses to disasters, developing in a therapeutic or a corrosive direction. Important factors in the nature of this development are the character of the disaster and how the affected community understands and evaluates it. According to this research, it is not so much what happened but why it happened that matters in making sense of a disaster [6–8]. Research has also found that the post-disaster work is extremely important for how affected people and communities develop and renegotiate their understanding of a disaster [9,10]. This understanding—including attributions of causality and blame—often has long-term consequences. It influences the post-disaster work as well as the development of preventive measures. It also influences the general level of trust

toward involved actors (not least governments and companies are associated with the disaster) [11]. In this sense, the understanding works performatively; beliefs and opinions—regardless of whether or not they are seen by others as valid—may have important political, social and material effects. It is important to note that how the disaster is understood may impact the social cohesion and identity of the affected community.

These findings constitute the starting point for this paper, which explores how communities make sense of a disaster. The focus is on post-disaster dynamics at the local level, in particular how a disaster influences local identity. A particular case is used to explore this—the largest wildfire in modern Swedish history—and interviews and a postal survey have been used to collect material on how the local population evaluates and make sense of the disaster. The aim is exploratory, in order to develop our understanding of the social effects of disasters at a community level. Guiding research questions are: (i) how do affected people interpret and evaluate the disaster, in particular its social effects on community cohesion; (ii) how can one explain their social interpretations; and (iii) what implications does this interpretation have for the wider issue of community vulnerability?

Exploring what is happening in a community that has experienced a disaster is of importance because environmental disasters cannot be fully prevented or mitigated, or even reliably predicted. It is therefore necessary to develop more robust societies and an important part of this work takes place on a community level. How the affected people interpret and evaluate a disaster and how the disaster influences their trust in extra-local actors as well as community members, have substantial effects on the community's social vulnerability, for example by affecting the local capacity for communication, collaboration and social reciprocity.

I will develop my argument in four steps. First, findings from disaster research are used to theoretically elaborate on community responses to disasters. Thereafter, the empirical case is presented—the largest forest fire in modern Swedish history—as well as the empirical material used to analyse this case. In the third step, the understanding of this disaster and its influence on the community's identity is investigated. Here a seemingly paradoxical finding emerges that seems to contradict some earlier findings in disaster research: although the disaster is seen as human-made, the affected community has engaged in very limited blame-making and responsibilization directed at involved parties and organizations. In the fourth step, I return to the earlier discussion on post-disaster dynamics and community identity, showing how the disaster, its social interpretation and the community identity dynamically interact and discussing the implications this has for community resilience.

## 2. Research Background: Extreme Events and Local Vulnerability

### 2.1. Framing Disasters

Crisis is often defined as a threat to fundamental structures or values in society, whereas disaster is seen as a manifestation and materialization of these threats that results in devastating and detrimental consequences [12,13]. Crises may lead to disasters and resources are often mobilized to prevent a crisis from developing into a disaster. What is important to note, however, is that the reverse development can also occur: the handling of a disaster may (unintentionally) lead to increased vulnerability and even new disasters [14]. How actors address the risks they face in the wake of a disaster is therefore central to understanding the social dynamics and trajectory of a disaster [15].

Not least when facing an unfamiliar situation, people and organizations have to develop a sense of what they are up against and what they need to do. By noticing, bracketing and labelling the details of a situation, people and organizations develop a plausible and coherent image that facilitates certain options and actions while constraining others [16–18]. The understanding of extreme events often takes the form of a narrative: a somewhat coherent storyline pulling together different events, actors and circumstances and establishing relationships between them [19–21]. A narrative tells not only *what* has happened but also *why* it has happened. This means that narratives not only contain

large amounts of condensed information and normative assumptions but also assign meaning to them, thereby directing attention and motivating action in certain directions [22]. Narratives are collective products, developed in social interactions, creating shared understandings and often resulting in unity between some actors and divergence from others who encompass other ways to narrate a disaster.

For example, how decision-makers, professionals and stakeholders narrate a disaster determines not only the post-disaster management but also what to learn from a disaster—what kinds of preventive measures and mitigation initiatives are needed to avoid similar disasters in the future. In this sense, narratives and social interpretations work performatively; all beliefs and interpretations—regardless of whether others see them as dubious—have consequences [23]. Thus, narratives, perceptions and interpretations cannot simply be reduced to cognitive or cultural phenomena. They influence actions and therefore have effects on the material, political and social worlds. Whether a particular environmental disaster is seen as exceptional or as something that may become a more regular occurrence has great implications for learning and the need to change practices and restructure institutions [24]. For example, beliefs about how to avoid future wildfires influence forestry practices and can transform forests and landscapes for decades to come [25,26]. Hence, how a disaster is narrated—what causes and consequences that are attached to it—is central for what kind of action that will be taken.

The main *cause* of a disaster may be located within nature (e.g., geophysical events such as earthquakes, droughts and flooding) or society (e.g., failures in technical systems such as power outages and industrial-chemical disasters). However, because a disaster always concerns the consequences of an event, the cause cannot exclusively lie within nature. Any particular society contains factors that enable a geophysical event to develop into a disaster. A disaster is thus never external to society, because it concerns the functioning of a system (or subsystem)—its vulnerabilities, robustness, resilience and adaptive capacity. Nevertheless, when people and organizations face a severe disaster, a central feature of their reaction is whether they ascribe the main cause to nature or society, with the latter meaning that a human decision was involved in making the disaster happen.

The *consequences* of a disaster can be of different kinds: economic, environmental, social, political and cultural. These consequences can also differ in both magnitude and range. Some extreme events are restricted to particular geographical areas, whereas others are associated with activities and phenomena that are geographically spread. The consequences of a disaster—whether stemming from nature or society—can be limited or widespread. Even types of disasters that cannot happen everywhere (such as floods, earthquakes and volcanic eruptions), may have consequences that are felt globally. For example, the 2010 volcanic eruption in Iceland resulted in 26 European countries issuing restrictions on flights. Furthermore, while any particular wildfire is spatially bound to a specific locality, it can at the same time be seen as a result of global warming, that is, be interpreted as an instance of a more general phenomenon of growing magnitude and frequency [27]. Lastly, consequences may be valued differently; as with many risk issues, disasters create both winners and losers [28]. Some persons or organizations may be financially ruined by a disaster, while others profit from it (not least due to market opportunities that emerge in its wake). To understand the varying responses to an extreme event, it is of central importance to investigate how it is understood and narrated, including what causes and consequences that are attached to it. The reason for this is that this understanding determines whether or not action is taken and what kind of action is deemed necessary [29,30]. Disasters therefore not only have practical and material consequences but also may change our interpretive paradigms [31,32]. They can change not only the way we think about them but also how we seek to manage them.

### 2.2. Therapeutic and Corrosive Communities

Since the 1980s, communities have been in focus in disaster research and disaster management approaches, largely as an alternative or a complement to top-down approaches [33]. The reason for this is that the local responses to disasters are heavily influenced by the local context [34,35]. An example

of this is the case of Hurricane Katrina, which hit New Orleans in 2005. Studies have shown that its severe consequences were amplified or weakened by social structures in the community. Many of the victims of Katrina were already dependent on societal support and many of the damaged houses were already dilapidated before being struck by the hurricane and flooded [36].

In this context, "community" often refers to residents within a spatially delimited area who recognize themselves as part of a social entity. Research has, however, been very critical when using the concept in an idealized and unproblematized way [33,37]. Community is often assumed to be characterized by cohesion, shared values and lack of conflicts (often with reference to Tönnies' concept of *Gemeinschaft*, despite the fact that his binary concepts *Gemeinschaft* and *Gesellschaft*—often translated as community-society—are analytical forms of social order [38]). Instead communities, like society at large, are characterized by homogeneity and heterogeneity, unity and diversity and cohesion and conflicts [33,39]. Despite its being a vague and multifarious concept, involving different meanings, it is increasingly referred to, both in disaster research and disaster management [33,40,41]. Community implies the existence of norms that at least partly guide the social life of a spatially delimited group whose members identify themselves as belonging to the socio-spatial group (though not necessarily in an exclusive sense; most people, probably all, have multiple belongings). This means that, despite all the differences and diversity, there is a sense of "we" in a community [39,40].

A stream of disaster research grounded in sociology or social-psychology stresses the importance of investigating and understanding how disasters influence the function and identity of affected communities. Obviously, many disasters have national implications and it is important to investigate how a particular disaster is evaluated nationally and how it influences national and international regulations. It is also important, however, to gain knowledge of its local impacts, not least because disasters always have distinct local consequences that may be overlooked if one only focuses on national aspects [34,35,42].

When a disaster hits a community, it often narrows the range of what people know about their situation (environmental as well as social) at the same time as they need to act in this environment in order to rescue themselves and protect their belongings [43]. Often they have to act in the new situation with incomplete and imperfect knowledge, or perhaps none at all, about what is happening and the best way to handle it. Issues of trust—who to listen to and whose advice to follow—are therefore brought to the fore. Research has found, however, that communities do not react and respond in a uniform way. Simply put, studies of the local implications of disasters find that communities follow one of two main lines of development: therapeutic or corrosive.

The term *therapeutic community* was originally coined by Fritz [44] to refer to a community hit by a disaster that not only maintained its social cohesion but was mobilizing its remaining resources to assist survivors. Also, rescue workers, neighbours, voluntary organizations and distant people were mobilizing to support the survivors and help to bring local residents together. In this way, positive emotions washed over the community soon after the disaster. Later studies show that people, including those considered vulnerable, respond in innovative and resilient ways that exhibit human creativity and responsibility [45,46]. A disaster provides opportunities for altruism, generosity and meaningful work. People may therefore come together rather than fall apart in the wake of disaster. Solnit [45] (p. 2) claims that "in the wake of an earthquake, a bombing, or a major storm, most people are altruistic, urgently engaged in caring for themselves and those around them, strangers and neighbours as well as friends and loved ones." A disaster can thereby serve as a doorway back into a community of positive social relations.

However, this is not the only story told about what happens in the wake of a disaster. A number of researchers have found that—rather than leading to the emergence of community solidarity—disasters can create social rupture. Blame, accusations, distrust and conflicts are often consequences of disasters, not least of technical disasters involving toxic exposure [6,47,48]. The concept of *contaminated communities* was invented to capture how people living in a toxic environment cope with their exposure and how their fear colours the life of the community [9]. Closely related to this concept is that of

*chronic technical disasters*, which is used for long-term exposure, which has distinct community effects that differ from the effects of natural disasters [49–51]. More recently, the concept of *volatile places* has been used to designate a situation where environments and human communities collide and public controversies emerge around this collision [52].

However, most relevant for this study is the concept of a *corrosive community*. This concerns an affected community's relation to its surroundings; that a community finds that governmental authorities, companies, media and other extra-local actors do not share its understanding of the situation—and may even question its understanding and experience—thereby failing to help the community to manage it. Empirically, the idea of contaminated community specifically concerns toxic exposure, where there is a need to draw a boundary around a polluted area. In contrast to this, corrosive community does not assume any particular character of the disaster. A corrosive community is also characterized by the deterioration of social relationships, not only external to the community but within it [47,53]. In a situation of confusion, stress, fear and anger, with competing definitions of what is at stake and who is responsible and with a multitude of (or no) proposals for the best way out of the situation, the community itself is put under great pressure. Not only can the soil and water be contaminated but also the social life of the community. Contamination can seep into the social fabric creating distrust of governments and corporations and may even foster social tensions within the community. Many disasters run silently through the structure of a community, creating winners and losers, splitting it into divisive fragments [6,48]. Also, responses from other social actors tend to amplify the social impacts. The affected community may see public agencies as foes rather than friends, because they are seen as unresponsive and as protecting bureaucratic prerogatives instead of assisting the local victims. In contrast to a therapeutic community, a corrosive community has not developed a shared narrative and identity but a divided one—one where the disaster has changed community members' sense of self, way of relating to others and view of society [6].

An often mentioned cause of why a therapeutic or a corrosive process begins is the character of the disaster; if the disaster has a human origin and includes toxic exposure, a community runs a higher risk of social corrosion. A disaster may have been horrifying but if survivors quickly see that it does not mean the end of the world and that the consequences may be severe but are known and manageable, then it is much easier for them to start to repair, rebuild and recover [47]. If the disaster involves toxic contamination, however, the survivors (and often also experts and public agencies) cannot grasp the character and severity of the threat and it is much harder to reconstruct the community both materially and socially. In those cases where the disaster is human caused, issues of responsibility, blame and trust also come to the fore. Much research has found that although natural disasters may have dramatic material consequences, it is technological disasters that have been the most disruptive in terms of psychological, economic, social and cultural impacts [47]. A major reason for this is that human causation implies expectations that the disaster should not have happened. Sociotechnical systems are legitimate because they are believed to be safe; otherwise they should not have been permitted. Natural disasters on the other hand are seen as "acts of God," not "acts of man." The issue of accountability—that an organization or individual shoulders responsibility for the disaster—is therefore central for creating therapeutic processes. Table 1 below sketches the main differences between therapeutic and corrosive communities.

**Table 1.** Therapeutic vs. corrosive communities.

|  | Therapeutic Community | Corrosive Community |
|---|---|---|
| Cause of disaster | Nature | Society |
| Biophysical and/or health consequences | Distinct | Ambiguous |
| Responsibility | Not central | Central |
| View of authorities and institutions | Trustworthy | Untrustworthy |
| Community consequences | Social cohesion | Social disintegration |

Thus, research on the social consequences of disasters, not least on the community level, stresses that it is not only the character of the disaster that matters but how it is acted upon by responsible organizations and how it is socially interpreted by the community. The sense-making around the disaster is crucial and in the wake of a disaster a discursive struggle often takes place concerning why it could happen and who or what to blame. When studying a disaster, it is therefore crucial to explore how it is socially interpreted, to ascertain what meanings—such as causes, consequences and responsibilities—are ascribed to the disaster.

## 3. Research Design

This study focuses on the social effects of a disaster for the affected community and in particular whether therapeutic or corrosive processes characterize the community consequences. This is done by investigating how the local residents interpret the disaster; both in terms of its causes and consequences and the performance of the organizations involved in fighting the fire and restoring the area. The material consists of interviews and a postal survey on views about the causes and consequences of the wildfire.

### 3.1. The Case: The Largest Forest Fire in Modern Swedish History

The summer of 2014 was extremely hot and it had not rained for several weeks (resulting in the highest forest fire weather index). When a local entrepreneur, hired by a large forest company, was performing subsoiling, a fire was ignited. The fire spread and two firefighting brigades (from different municipalities) fought it separately. Initially, the fire seemed to be under control but strong winds and the extremely dry vegetation caused the fire to go completely out of control. During the course of a single day it spread from 2800 to 13,000 hectares, causing it to be seen as a major crisis by the authorities and leading to a reorganization of the operations. Not only governmental agencies, fire and rescue services from other parts of Sweden were involved but also firefighting aircraft from France and Italy that water-bombed the fire. In total, 2300 people (including volunteers) were mobilized for the firefighting efforts (which included evacuating some 1000 people and 1700 animals). Eight days after it began, the wildfire was brought under control (most of the above-ground fire having been put out) and five weeks later (11 September) it was formally declared extinguished.

The damage was considerable. The fire caused one fatality, almost 100 buildings were damaged or destroyed and 1.4 million cubic meters of timber were damaged. In total 15,000 hectares were burned including three nature reserves and some ten areas for habitat protection and nature conservation. The total cost of the fire is estimated to at least 100 million euros [24]. The burned area was owned by a handful of large forest companies and forest associations and some 100 small-scale private owners. Later, a nature reserve was established covering almost half of the affected area (6400 ha). The Swedish state offered either economic compensation or a new forest property in the vicinity for those owning forest in the newly established nature reserve. From an international perspective, it was a rather small wildfire. However disasters cannot simply be defined in terms of biophysical consequences (e.g., size of burned area), economic damage (e.g., drop in property values and costs for fighting the fire), or health consequences (e.g., deaths and serious injuries) but are much more contextual; they also need to be framed and recognized as extreme by the community or the wider society. Also, a disaster should not be defined only from an international perspective but from a national and local perspective as well. Thus, even if this wildfire was small when viewed globally, it was nevertheless the largest fire in modern Swedish history; it strained the country's fire-fighting and crises management services and resulted in a number of national evaluations of Sweden's capacity for fighting forest fires and crisis management more generally. In this sense, this disaster became a formative moment in Sweden's understanding of and preparedness for wildfires.

### 3.2. Materials and Methods

This study is based on three different data collections, all made in the wake of the disaster.

*Interview Study 1* uses a key informant approach. Selection criteria were that informants (i) represented organizations that were involved in managing the forest fire and its aftermath; and (ii) were employed by organizations affected by the fire. The study consists of 10 interviews with state-employed forest consultants (3), representatives of forest companies (3), private forest associations (2) and the farmers' association (2). The interviews were conducted 2–5 months after the fire (October 2014 to January 2015). The interviews included questions about their perceptions of the cause of the fire, the actions taken during and after the fire, the consequences of the fire, the risk of large-scale wildfires in the future and what can be learned from this wildfire.

*Interview Study 2* uses a key informant approach, where the selection criteria were that participants were deeply involved in the post-disaster work and were presumed to know about not only their own organization's viewpoints but also how others (organizations involved in the post-fire work or people directly affected by the fire) evaluated the fire disaster. The study consists of interviews with 20 persons representing organizations and/or interests deeply involved in the post-fire work. These were governmental agencies (5 respondents), forest companies (3), forest associations (2), individual forest owners (3), non-profit NGOs (4), insurance companies (1), residents/house owners (2). The interviews were conducted 8–9 months after the fire (April to May 2015). The interviews included questions about what happened in the aftermath of the fire, how different organizations and individuals assess this work and how the perceived legitimacy of and trust in, involved organizations were affected by the fire.

The interviews were conducted in Swedish and were recorded and transcribed verbatim. Since the interviews were semi-structured, they were in general open, allowing the interviewer and respondent to examine new ideas that were brought up during the interview [54,55]. NVivo software for the analysis of qualitative data was used for conducting a contextualized thematic analysis [56]. Of the 24 themes constructed, three were deemed relevant for this study: trust, social cohesion and disagreements and conflicts. (Example of other themes, not relevant for this article, were biological diversity, climate change, economy, information during the fire, prevention work, replantation and the evolution of the fire disaster.)

A *postal survey* was sent to all people directly affected by the wildfire. Directly affected persons were defined as: forest owners, permanent residents, summer cottagers and evacuees (both those who in fact were evacuated and those asked by public authorities to be prepared to evacuate). The survey was distributed in March 2015 (7 months after the fire) by Statistics Sweden to all adults (18 years or older) and the participants were guaranteed confidentiality. No compensation was given to the participants. This sample consisted of 960 individuals and, after two reminders by letter, the response rate was 78% (746 respondents). The survey included a total of 133 questions. The themes covered in the survey concerned both general views on forest fires in Sweden and views on the particular wildfire in 2014 (such as causes and consequences of the fire; evaluation of involved organizations' work during the fire; and if their trust in these organizations as well in the local community has changed because of the fire). A descriptive and bivariate analysis was done using SPSS software.

Judging the reliability of post facto description is complicated but what this study aims to investigate is social interpretations of the fire—how the wildfire was understood and made sense of by the affected people and its consequences for the community's social cohesion.

## 4. Results

The analysis is structured in two parts, the first centred around what social effects the respondents ascribe to the wildfire, in particular how it affected the community identity; and the second around what causes respondents claim led to these effects.

### 4.1. What Were the Effects on Social Cohesion?

The survey shows that those who were affected by the fire believe that social cohesion in the area has increased (see Table 2). The survey was made nine months after the fire disaster, which means

that the respondents should no longer have been caught up in their experiences from the acute phase, when the entire community mobilized to combat the fire and minimize its consequences.

**Table 2.** After the fire, cohesion has increased between neighbours in the affected area.

| Statement | Count | Percent | Accum. Percent |
| --- | --- | --- | --- |
| Agree | 339 | 45.4% | 57% |
| Partly agree | 158 | 21.2% | 26.6% |
| Disagree | 98 | 13.1% | 16.5% |
| Don't know or No answer | 151 | 20.2% | |
| Total | 746 | 100% | |

Obviously, self-reported changes alone do not provide a robust enough basis for drawing firm conclusions. However, this view is strongly supported by the interviews with professional representatives of organizations that worked in the area during and after the wildfire. Several of these describe having been amazed by the level of community engagement in supporting both the firefighting efforts and their neighbours. A forest professional, working for a large forest owner, describes it in the following way:

> I also think that the positive thing is, this feeling, a feeling of "we" in the entire community. At least during the fire, it was amazing ... That's probably what struck me the most, that everyone helped out (15 December 2014).

A representative from the Federation of Swedish Family Forest Owners (LRF Forestry) says:

> At the local level, I have to say it was incredible, if you look at the resources, volunteer resources that were out there. If you were out driving, there were farmers who joined together. I visited a farm and they had like put out beds, there was food, all the tractors stood there, they really joined together and helped out. So I must say this was fantastic (26 May 2015).

Likewise, a representative from the County Administrative Board talks about "the amazing community rallying around the wildfire" (10 April 2015). A volunteer from the Voluntary Resource Group (which emerged in response to the wildfire) says "There were so many people who wanted to help that we had to say no" (18 May 2015).

Worth noting is that this feeling persists, according to both the community residents and the professionals working in the area. A cottager describes it, nine months after the wildfire, in the following words:

> It's like the whole community says that there's a new spirit in the whole community, we talk to each other more ... People talk to each other in a different way. So that's been good. At the very least, the generosity of the people around one, that's amazing (cottager, 29 April 2015).

In a similar vein, a forest owner living in the area says:

> It has changed a hell of a lot, even between neighbours. There was a hell of a surge of support in the village, you could say, you'd never have believed it (private forest owner, 15 April 2015).

Most vividly, a regional representative of the Federation of Swedish Family Forest Owners (LRF Forestry) describes this change:

> But then, something happens, I don't know, it's hard to explain. For the cohesion of the community and its villages. Someone wrote a letter to the editor: this is the grumpiest community in Sweden, what's happening? How can people start talking to each other?



We have farms in the villages where they've been feuding for three generations. Now they went and made firebreaks together, sat and had coffee, they gave some gibes to each other but still they talked to each other ... So, that's what I think is the most fascinating thing about all this. I tell people, if they ask, that if they're enemies you damn well need a wildfire so that you can get along (9 January 2015).

The survey results (see Table 3) also support this, showing that those who were affected by the fire also believe that the increased cohesion (in terms of taking care of each other) will last.

**Table 3.** Forest fires will cause residents in the affected area to care more about each other.

| Statement | Count | Percent | Accum. Percent |
|---|---|---|---|
| Agree | 321 | 43.0% | 50.2% |
| Partly agree | 204 | 27.3% | 31.9% |
| Disagree | 115 | 15.4% | 18.0% |
| Don't know or No answer | 106 | 14.2% | |
| Total | 746 | 100% | |

This should not be allowed to overshadow the fact that there was also some irritation. Respondents mention that afterwards some forest owners felled more trees than they needed to (i.e., undamaged trees that were felled for economic reasons); that the police's roadblocks sometimes hindered residents and forest owners from supporting firefighters; that fire tourists drove randomly on small roads, taking photos of people and places; and that some politicians seemed to visit the place simply for the publicity (the national and local elections took place a month after the disaster).

Also, some consider the nature reserve to have been made too large, implying that some villages will have unnecessary traffic on their roads.

Furthermore, the disaster was not without personal consequences. The survey asks about personal losses caused by the disaster. Even if the most common answer is that the fire has not caused them any personal loss at all, the respondents mentioned a number of consequences (see Table 4).

**Table 4.** What is the greatest loss for you personally as a result of the fire? Source: the table was originally published in Reference [24].

| Statement | Percent | Count | Accum. Percent |
|---|---|---|---|
| The forest fire has not personally caused me any losses | 29.2% | 218 | 29.2% |
| I am not able to pick mushrooms or berries anymore, or to fish or hunt in the area | 18.2% | 135 | 47.4% |
| The opportunity to have experiences close to nature is gone | 15.4% | 115 | 62.8% |
| The burned forest is horrible to look at/the forest is gone | 11.8% | 88 | 74.6% |
| Worry about new fires/Psychological problems/PTSD/Grief/Lasting sense of insecurity | 10.8% | 81 | 85.4% |
| Loss of economic value | 6.0% | 45 | 91.4% |
| Other | 2.7% | 20 | 94.1% |
| Don't know/No answer | 5.9% | 44 | 100% |
| Total | 100% | 746 | |

An important point is that very few state that their greatest personal loss concerned their economy, property or health.

*4.2. Perceived Causes of Social Cohesion*

As emphasized above, a community hit by a disaster may develop in a corrosive or a therapeutic direction. In this particular case, the affected people state that the wildfire has strengthened the social cohesion of the community and the question emerges: How do they understand this development? What made the community develop stronger social cohesion instead of falling apart? The interview

guide did not include any questions about social cohesion, only on general effects (ecological, economic and social) of the disaster. However, in answering these questions many respondents included descriptions of increased social cohesion as a positive effect of the wildfire.

A common starting-point for interviewees' descriptions is that many local residents had an extremely pressing situation, both during and after the wildfire—not knowing when it was safe to walk in the area (due to damaged trees that can randomly fall)—and also that many individual forest owners had found it very hard to really see what had happened with their forest. As a response to this situation, care-taking emerged, both spontaneous and organized (see [57] for a description of the voluntary work and its organization). A forest professional said that:

> I only had one objective from the start, it was that no one should kill themselves. We had two concerts at the church and the last thing I said was "if you don't have anything else to do, take some coffee and pastries and visit those who you don't see during the days, because they're the ones who are at risk." Sitting in the bush alone and pondering, unable to think straight and who don't say a word, because they're the ones who, they are at greatest risk. Those who talk a lot are at lower risk (representative of the Federation of Swedish Family Forest Owners, 9 January 2015).

Representatives from forest companies, forest owners and the forest agency similarly describe that in the acute phase they supported the local forest owners as much as possible. A representative of the Forest Agency working in the area describes it in the following words: "to buy timber and try to compete, it doesn't feel relevant, here you help your supplier to get the timber out and that's it. It's not about doing business but more about getting it done." (16 October 2014).

Thus, the extreme situation seems to have triggered different forms of caring actions, implying a strengthening of social relations and even the creation of new ones. As two forest professionals describe it:

> The willingness to volunteer during the fire has meant that neighbourliness has increased and that you care more about each other and some old conflicts have almost been resolved, too, because, well, you've had no other alternative than to help each other and then perhaps you realized that you only benefit from, well, helping each other (representative of the Federation of Swedish Family Forest Owners, 20 January 2015).

> Thus, you can see a positive social thing, that the landowners who are in this area, they have gotten a completely different point of contact, that they've become much more sociable with each other, they meet and talk and it's like, you need to talk again and again. And at the evenings that we have organized around the theme of forest regeneration we see that there are a lot of other issues that come up also . . . So it's a social thing, there's more social coherence in the community, so to speak, you can keep an eye on each other better (regional representative of the Forest Agency, 16 October 2014).

## 5. Discussion: Why Was There No Social Corrosion?

The interviews and the survey reveal a rather homogeneous view of the wildfire and its social effects on the community, articulated by both the local residents and professionals working in the area during and after the fire. Obviously, the respondents bring up different things in the interviews and emphasize certain events and themes differently. However there are no respondents who claim that the disaster has had severe social effects in terms of increased controversy or antagonism within the local population. Most of the respondents instead claim that the community has been substantially strengthened by their experience of the disaster. This result is interesting not least because the wildfire was human caused, making it more likely that issues of accountability and processes of blame-making would come to the fore [58]. A human-made disaster, which in the acute phase threatened what humans valued, should be followed by strong criticism of those who caused the disaster or who failed

to contain the wildfire in the early stages and limit its consequences. Even if the disaster later turned out to have had less severe consequences than originally believed, strong criticism could nevertheless be directed at responsible organizations. However in this case, it resulted neither in strong criticism of external others, nor in internal tensions. Instead, there seems to be a rather understanding attitude toward responsible organizations and a mainly therapeutic process at work in the community; or at least, when the affected parties look back on their experience and narrate it, they stress how the community was welded together, both during the acute phase and after the fire was extinguished.

How to explain this? I will first discuss how this particular view of the disaster could develop and take hold in the community and then how it relates to changes in community identity. By way of conclusion, I will raise the issue of social vulnerability and the extent to which this kind of experience—an environmental disaster resulting in increased local cohesion—implies that the community has become more resilient.

### 5.1. What Is the Reason for This Positive Evaluation of the Disaster?

As stressed in Section 2, a community's response to a disaster can develop in therapeutic or corrosive directions, depending on the character of the disaster, or rather its perceived character. How a disaster is understood in terms of causes and consequences and how organizations handle the disastrous situation, influence the affected community. Obviously, the perception, sense-making and narration of a disaster do not take place in a social and material vacuum. The starting point is the extreme event, which is then interpreted and this interpretation is dependent on an actor's interest and position (social as well as spatial) as well as what stories and social interpretations the actor encounters (through traditional media, social media, rumours, small talk, etc.) [28]. I wish to stress five important characteristics in particular that influenced the social interpretation of the wildfire disaster.

First and most importantly, the disaster does not seem to have resulted in any severe long-term consequences for the community. Among the major consequences of a wildfire disaster are loss of life, destruction of property and economic losses [59,60]. Even if the disaster was dramatic and disastrous in its acute phase, there are few long-term consequences attached to it. Apart from one fatality and one serious injury, there were few health consequences, either direct (caused by the wildfire) or indirect (caused by the crisis management, for instance the evacuation) and there was no uncertainty in the community about long-term health consequences of the wildfire. Furthermore, there was little property loss, mainly because the area was sparsely populated. Only a few of the damaged buildings were permanent residences. Most of the forest owners had insurance, which means that they received economic compensation for the burned forest. Thus, for those living or owning property in the area, the consequences were rather limited.

Secondly, no social stigma [61] was attached to the locality. In cases of contamination, places may come to be associated with a negative meanings and characteristics. In this case, there was no toxic component involved and there was no such ground for stigmatization. In fact, the disaster seems rather to have associated the area with positive meanings. Media reported that a nature reserve was to be established on about half of the burned area (6400 ha), with three entrances, marked trails, resting places, a barbeque area and a lookout tower. There are regular excursions and organized tours to the area. Thus, a rather remote and not densely populated locality suddenly became well-known, attracting visitors such as people who are working with or interested in forestry, biological diversity and nature conservation, as well as tourists interested in visiting the site of the disaster or the nature reserve. Currently, 14,000 cars travel to the area each year and the lookout tower (established in Autumn 2017) attracted 36,000 visitors during its first year in operation.

Thirdly, little responsibilization took place in the aftermath of the disaster. A common finding in research is that, after a disaster, strong criticism in form of blaming and responsibilization is often directed at organizations believed to be responsible for a disaster [62]. This is not least the case if the disaster is seen as human made. The public investigations after the wildfire all place responsibility for the wildfire on the forest sector and the crisis management. The investigations stress that the

forest sector needs to develop active measures to prevent forest fires in the future [63], the emergency services and fire departments need to better incorporate existing knowledge of forest fires into their training and routines [64] and crisis management needs to clarify the responsibilities and improve its coordination of different actors [65]. However, in contrast to these investigations, those directly affected (local residents, summer cottagers and forest owners) do not emphasize issues of accountability in their evaluation of the forest fire. Although most respondents state that the wildfire was human caused and could have been avoided, this is not followed by any critical claims about the organizations that caused the wildfire or failed to prevent it from becoming so severe (mainly a forest company but also crisis management). A reasonable explanation for this is that the few negative consequences of the fire for the community made respondents less inclined to find scapegoats for the disaster; they were just happy that the wildfire did not result in worse consequences.

Fourthly, the disaster did not lead to any litigation process that involved the community. Much disaster research suggests that this kind of process often tears a community apart [66–69]. Most property owners were compensated by their insurance companies, which means that they did not need to go to civil court to receive economic compensation. Instead, the national insurance companies are now are pursuing a litigation process against the forest company that caused the fire, demanding 25 M euros (277 M SEK) to cover their payments to affected forest owners.

Fifthly, most economic costs were allocated to non-local levels and in this sense were invisible from a community perspective. Many forest owners had fire insurance covering much of the costs for the burned or damaged forest. (Note, however, that 45 respondents state that their greatest loss was economic, probably because they lacked insurance or only were partly compensated by the insurance.) Those owning property in the newly established nature reserve in the area were either economically compensated or offered a new forest property in the vicinity. The state and the municipalities covered the costs for the emergency operations (fire-fighting and crisis management). The total cost for the fire (the emergency operations and the damages) is estimated to at least 100 million euros. Thus, extra-local actors covered most of the economic costs of the wildfire and the issue of the economic cost of the wildfire was almost invisible in the public discussion.

## 5.2. Why Does Everyone Tell the Same Story?

So far, the local view of the forest fire and its aftermath have been in focus. What about its effects on the community and its identity? The interview study shows a rather strong monolithic narrative in which almost all interviewees—local residents as well as professionals working in the area during and/or after the fire—point to a substantial change in the social cohesion in the community.

Narratives are a kind of story-telling that communities or networks use in attempting to deal with specific problems collectively [70,71]. It gives a historical account of a problem (including its causes and consequences) which motivates, guides and legitimizes decisions and actions. This collective attribution of meaning is based on shared experiences and interactions.

The experience of a disaster is maintained and developed through stories told about it, the construction of strong narratives guiding our evaluation—cognitive, normative and emotive—of the disaster and its effects. If successfully embraced, a narrative is naturalized—meaning that it becomes taken for granted. In this case, the narrative concerned not only the wildfire but also the community, how it was hit by the wildfire and how it engaged in both supporting the firefighting and helping community members. It is a story of care, support and altruistic action, where community members as well as professionals working in the area helped residents. It is a story much like that which Solnit [45] finds may take place in the wake of disaster. The spatial anchoring of the narrative—it concerns a particular place and group of people—also makes the community identity an important part of the narrative.

There is a dynamic relation between experience, narrative and identity, which influence each other in reciprocal ways. An event may have causal effects but they are mediated through our social interpretation of the event—how we make sense of it. Likewise, a narrative has also causal effects but

this narrative is not created in a social and material vacuum. The community identity influences how a disaster is narrated but is also affected by the disaster, as well as by how it is acted upon by local and extra-local actors. Thus, in this case the shared experience also had implications for the shared identity. How we talk about an event (including what took place in its aftermath) is an important factor in how we understand both a disaster and ourselves as the affected group.

A community identity is often strengthened (and sometimes even constructed) by an external enemy, for instance through accusing an organization or a person of being responsible for the disaster. Furthermore, if extra-local actors—mainly in the form of public agencies, companies and media—do not understand how the disaster is viewed by those affected, this may not only deepen the conflict but also increase the cohesion of the community [72]. The phenomenon "outsiders do not understand" is common in contaminated communities where extra-local actors' descriptions of what has taken place differ significantly from those of persons locally exposed to the disaster [73]. As mentioned above, there was rather little responsibilization in this case and the interviews and survey show that the affected people are mainly positive toward the external actors and their work. There is, however, an interesting exception to this. When asked about tensions and conflicts in the recovery work, many interviewees mentioned a debate article in a Swedish daily paper shortly after the wildfire, where 21 professors proposed that the area should become a nature reserve, arguing that the wildfire provided a historic opportunity to have a large, cohesive forest area with the special qualities that the fire caused and at a low cost [74]. The interviewees describe how this proposal caused a storm of irritation in the community; many local forest owners felt that these forest researchers were arrogant and did not understand that the wildfire was a tragedy for them. An interviewee (a forest owner) spoke of the "heartless professors," called them "muppets" and says, "it makes you wonder what they were thinking" (21 April 2015). Even if this can be considered an exception, it nevertheless is so frequently mentioned in the interviews that it is reasonable to treat it as an important part of the narrative and as strengthening the community identity. It does so not so much by being about an external enemy but by being a funny story to tell; for as sociological research has found, good stories, jokes and pieces of gossip are important for creating and strengthening group identity [75].

As shown above (Section 5.1), the wildfire is positively evaluated by the community and the contrast between how it was perceived in its acute phase (threatening human life and property) and after the recovery phase (having mainly positive consequences) is remarkable. To summarize the results from the interview studies, the wildfire is narrated as exceptional (caused by operational failure and extreme weather), having few severe long-term consequences and many positive ones and involving no ambiguities or uncertainties concerning future harms. This gives little cause for externalization of blame and responsibilization and also gives reasons for strengthening the community identity. There were few losers and after a rather short period of time everyday life and daily routines were restored, though now with the additional positive experience of having been able to manage the disaster. The disaster seems to function as an source of cohesion by (i) providing a shared situation (being hit by a disaster) and (ii) a shared focus (rescue and recovery); and these may (iii) lead to a transcendence, or least a toning down, of social differences and group belongings (shared identity). Afterwards, when the disaster was shown to have few negative consequences (at least locally), there were few reasons to blame anyone. Thus, this is a story of a severely threatened community that ultimately was not damaged in any substantial sense and instead came out of the experience socially stronger in terms of finding that its members were united through their efforts to successfully combat the disaster and its consequences.

Whereas studies have shown that disruption of a landscape may result in a new identity, bound up with the disaster and with memories of a lost landscape [76] as well as lost emotional ties to the landscape [77], in this case it seems rather to have led to strengthened identity. People who formerly did not have so much in common were united through an exceptional experience that included not only the wildfire and the burned landscape but also their becoming a community characterized by trust, involvement and care-taking.

I propose the concept *invented community* to capture what has taken place in our case. Within the social sciences, the notion of "imagined community" [78] is well-known; this is an idea of community that has institutional power and also can change the community it purports to represent. Nationalism is the best example of this, where people who are spread over a large geographical area and never have met start to define themselves as belonging to the same entity and often also experience a deeply felt horizontal comradeship [78]. In contrast to the concept of imagined community, invented community refers to a community which is close to the traditional one: involving direct and frequent interactions between persons populating a limited geographical area [79] who share a history and connections to each other [80]. However it is also an *invented* community in the sense that people talk about it in new ways because of the wildfire; "the disaster brought us together in a new way, it amalgamated us and a new community identity was developed." A social narrative is created—not only about the disaster and the recovery work but also about the community and its character. In this sense it is invented, as it provides a particular way to talk about and interpret the character of the community. Because a community is partly a self-construct—it ceases to exist if its members do not see themselves as belonging to it or do not interact with each other—this means that social narratives are not external to the community identity but part of it. In contrast to ecological and economic consequences, specific social consequences—such as those concerning self-identity and trust—are deeply affected by community members' perceptions of it. This does not mean that the binary concepts of therapeutic and corrosive community are obsolete but only that the character of the community is not solely caused by exogenous factors. The internal social interpretation of the event and its consequences also matters. The disaster, how it is socially interpreted and the community identity are all dynamically related. This also means that the community identity may be of a transient character. Regardless of whether it has developed in a corrosive or therapeutic direction, it may change over time.

Some limitations of this study need to be acknowledged. First and foremost, judging the reliability of post facto declarations is notoriously complicated. Obviously, changes in community identity are better validated if other sources than self-reported changes are also used. Nevertheless, I find it justifiable to claim that the community identity has changed because the interviewees included not only local residents but also professionals working in the area during the recovery phase. Also, as stressed in the theoretical approach, self-reported changes matter when it comes to the issue of identity. The reason for this, as stressed in the discussion above, is that social interpretations of an event and narratives about a community identity are also a part of this identity.

Another and more serious limitation concerns temporality. Even if the interview studies were made with a six month interval between them and the survey was conducted about eight months after the disaster, it is hard to gauge the long-term effects on community identity. Thus, this study cannot give any answer about whether the shared experience and the social narrative will have lasting effects on the community identity. The social effects of the wildfire may result in a partly changed community identity that will persist, or a community identity that eventually will return to where it was before the wildfire. The design of the study does not therefore allow any conclusions to be drawn about the lasting effects of the wildfire on community identity.

A third limitation concerns the possibility for drawing more general conclusion from the case. There exists no scientific knowledge about the community and its identity before the disaster, which means that it is not possible to know how much its original character (pre-existing conditions of social, economic, political and cultural character) caused this result; this kind of invented community. Consequences are never shaped in a social vacuum and it is hard to know what would have happened if another community had been hit by the forest fire disaster. Furthermore, communities are always embedded in a wider context, consisting of assemblages of actors, networks and cultures [81]. It was not, however, within the scope of this article to investigate exogenous factors that may influence the internal dynamics of the community. These include more general societal change (e.g., urbanization and economic restructuring which deeply affect rural areas), environmental changes (e.g., climate change) and national regulatory frameworks (including forest practices and crisis management.

*5.3. Conclusion: Invented Communities and Social Vulnerability*

To what extent has the experience of the wildfire strengthened the capacity of the community to handle similar events in the future? This question is not least of relevance in the current discussion on how to adapt to climate change. There is currently a challenge of global dimensions to decrease vulnerability and increase the robustness of societies in order to minimize the adverse effects of climate-related extreme events [1]. Even if extreme events cannot fully be prevented, or even reliably predicted, it is possible to develop more resilient societies. Apart from their global and regional aspects, vulnerability and resilience also have distinct local characters. Communities in the same region may be differently vulnerable—as can also different groups within a community, due to inequalities [33,36].

In this case, the community seems to have developed in the direction of probably being more resilient to new extreme events, including other ones than wildfires. It now has positive experiences of social mobilization and voluntary engagement. The social narrative and community identity, which stressed that the community is characterized by a spirit of care and trust, will also facilitate local collaborations and altruistic engagement. In this sense, the wildfire has resulted in a less vulnerable community.

Alongside these positive effects, however, there may also be negative ones. In particular, I would like to stress the absence of local calls for structural change to reduce the risk of future large-scale wildfires. As the interviews and survey show, there are rather few critical voices in the community asserting the need for structural and organizational changes. The strong, almost monolithic, narrative that has taken hold in the community also implies that there is little space for alternative stories or claims that are not easily adapted to this narrative. The public investigation heavily stressed organizational failures—particularly by the forest sector and crises management authorities—as contributing to the wildfire and to its having grown out of control [63–65,82]. They stress the need for systematic preventive work (including risk and vulnerability analysis), organizational change and clarification of responsibilities and even have considered the option of prohibiting certain forest practices in cases of extreme weather—in contrast to the opinions voiced by the community residents. In this sense, there is a paradoxical situation: through the wildfire, the community has been made better prepared to handle similar extreme events but at the same time it does not claim that there is a need for more structural changes to reduce the future risk of such events.

To sum up, there is a dynamic interplay between a disaster, its social interpretation and the identity of the affected community. As this study shows, it is crucial to understand how a disaster is perceived and interpreted.

**Funding:** This research was funded by the Swedish Research Council Formas, project Risk governance, legitimacy and social learning in the handling of the forest fire in Västmanland [211-2014-1875].

**Acknowledgments:** I would like to thank Viktor Hedermo and Daniel Sjödin (both at Örebro University) for conducting parts of the interviews. Daniel Sjödin and I co-constructed the survey and Daniel performed the data analysis and prepared Tables 2–4. I also thank the four anonymous reviewers for constructive comments.

**Conflicts of Interest:** The author declares no conflict of interest.

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
