# Peer review of "Invented Communities and Social Vulnerability: The Local Post-Disaster Dynamics of Extreme Environmental Events"

_sustainability, doi:10.3390/su10124457_

Round 1
Reviewer 1 Report
An inspiring paper which, elaborating on the idea of 'invented communities', will stimulate debate on linkages between disasters and social cohesion.
I do have a few requests though, and some comments:
Since 'community' features so strongly in your paper, it might be worthwhile to take a more critical or differentiated view on community concepts. May I recommend you take a look at a very recent publication by Titz et al. 2018, Uncovering Community, published in Societies, Vol. 8 Issue 3 ? You could refer to contested notions of 'community' both in your introductory chapter and later, when you discuss your results. In fact, I have the feeling that there is a certain flippancy with which you use the term community. Later in the text, you do show discrepencies within so-called communities, so why not find a more appropriate term?
In this vein, line 31, for instance, reads as if 'community' was a given, and line 41 "...people and communities develop and renegotiate their understanding of..." as if there was always ONE common understanding. One might argue that this is usually not the case, and indeed, you do show that later in your text.
Lines 33-35: a good and promising start of the paper!
Lines 103-105 on narratives: an important observation, thank you!
Lines 131/132: "...Lastly, consequences may be valued differently; as with many risk issues, disasters create both winners and losers." Exactly. This is one example showing that 'cohesion' or 'common understanding' is not what you necessarily find when you work with people at the local level, in your case with those affected by the wildfire.
Chapter 2.2: Interesting!
Lines 2017-208: "...and it is much harder to reconstruct the community..." - Is it possible at all to reconstruct something that has probably not been there in the first place? It is usually extremely difficult, in retrospect, to find out what exactly constituted a group of people (cohesion, belonging,...) before it was destructed.
Lines 294-295: For context, maybe elaborate a little bit more on the other themes.
Table 2: 20 % did not know or did not answer. Probably because the term 'cohesion' is too difficult/complex to be used in an interview/questionnaire? Just a thought.
Line 328, direct quote: I wonder whether the interviews were conducted in English or Swedish? If the latter, what term did they use that was then translated into 'community'?
Chapter 5: It becomes quite clear here that by community, neighbours or local residents are meant. Maybe say so instead of using the blurry term 'community'? Since 'community' is your key term in your argument, chapter 5.2 might be a good section to (again) clarify whom exactly you mean: All immediate neighbours? All residents of the area?... And what about those 15 or so percent who disagreed with the idea of increased cohesion?
I feel that if you could address some of the issues mentioned here (and I believe this can be done quickly) your paper (which is already strong!) could gain some rigour.
Author Response
Review 1
An inspiring paper which, elaborating on the idea of 'invented communities', will stimulate debate on linkages between disasters and social cohesion. I do have a few requests though, and some comments:
Since 'community' features so strongly in your paper, it might be worthwhile to take a more critical or differentiated view on community concepts. May I recommend you take a look at a very recent publication by Titz et al. 2018, Uncovering Community, published in Societies, Vol. 8 Issue 3 ? You could refer to contested notions of 'community' both in your introductory chapter and later, when you discuss your results. In fact, I have the feeling that there is a certain flippancy with which you use the term community. Later in the text, you do show discrepencies within so-called communities, so why not find a more appropriate term?
In this vein, line 31, for instance, reads as if 'community' was a given, and line 41 "...people and communities develop and renegotiate their understanding of..." as if there was always ONE common understanding. One might argue that this is usually not the case, and indeed, you do show that later in your text.
Very important point: I have written a new paragraph in section 2.2 which problematize the concept and briefly come back to this in the conclusion (section 5.2)
Lines 33-35: a good and promising start of the paper!
Lines 103-105 on narratives: an important observation, thank you!
Lines 131/132: "...Lastly, consequences may be valued differently; as with many risk issues, disasters create both winners and losers." Exactly. This is one example showing that 'cohesion' or 'common understanding' is not what you necessarily find when you work with people at the local level, in your case with those affected by the wildfire.
I agree, but as I see it, this makes the empirical result even more interesting. My original thought were that it should be a more differentiated view, but I did not find it. And my explanation (section 5.1) is that the disaster creates very few losers at local level. Please also see my comments above (that I in the revision included a more problematized view on “community”)
Chapter 2.2: Interesting!
Lines 2017-208: "...and it is much harder to reconstruct the community..." - Is it possible at all to reconstruct something that has probably not been there in the first place? It is usually extremely difficult, in retrospect, to find out what exactly constituted a group of people (cohesion, belonging,...) before it was destructed.
Good point: I have not explicitly discuss it here (line 207-8) but included in section 2.2 a new paragraph that stress that community is not necessarily associated with social cohesion, and in conclusion (section 5.2) stated that a substantial limitation of this study is that there is no systematized knowledge about the community’s character before the disaster hit it (i.e. I have only information about how the community is narrated after being hit by the forest fire). At the same time, I have better stressed the importance of how a disaster is narrated and also how the community identity is narrated.
Lines 294-295: For context, maybe elaborate a little bit more on the other themes.
Done
Table 2: 20 % did not know or did not answer. Probably because the term 'cohesion' is too difficult/complex to be used in an interview/questionnaire? Just a thought.
My experience is that cohesion (Swedish: sammanhĂĄllning) are easily understood by most Swedes (of course roughly, but it is a common notion used in the Swedish vocabulary).
Line 328, direct quote: I wonder whether the interviews were conducted in English or Swedish? If the latter, what term did they use that was then translated into 'community'?
Good point, but it is hard to explain the Swedish which terms are used by the interviewees. In this quotation, a notion close to the German Heimat is used ((Swedish “bygd”), which is stronger than community. I hope that the more problematized view of community, which I have included in this version, implies that the reader is aware that the conceptual usage may differ between the interviewees. Additionally, I have included information (3.2) that the interviews were made in Swedish
Chapter 5: It becomes quite clear here that by community, neighbours or local residents are meant. Maybe say so instead of using the blurry term 'community'? Since 'community' is your key term in your argument, chapter 5.2 might be a good section to (again) clarify whom exactly you mean: All immediate neighbours? All residents of the area?... And what about those 15 or so percent who disagreed with the idea of increased cohesion?
Agree: As informed above, I have included a more qualified discussion on the concept community (section 2.2) and come back to it in section 5.2. Hopefully it shows that the multiple conceptual usage of “community”. At the same time, it is, to my opinion, wrong to reduce community to residents and neighbors. Community, as discussed in section 2.2., includes a sense of belonging to a place and a group. And the social narrative that I have found through my analysis strengthen this.
About reasons behind that about the 15% that disagree that the fire will not lead the residents to care more about each other. I have not included any discussion of it in the paper, this because it is hard to give reasons behind this. In the survey, there is no pattern in terms of gender, age or years living in the community that correlates with this response.
I feel that if you could address some of the issues mentioned here (and I believe this can be done quickly) your paper (which is already strong!) could gain some rigour.
Reviewer 2 Report
I found this to be a novel, valuable and well written application of social-construction and sensemaking theory to understanding the consequences of disasters. This is an underdeveloped area in disaster research and the authors make an important contribution.
One question I had was the extent to which a largely homogenous community may predispose the results toward a positive outcome, in contrast to not only more diverse communities, but especially communities where barriers exist between ethnic and cultural sub-groups in the community thus limiting the development of a shared narrative. This community may not have been meaningfully at risk of a more corrosive response but reflect another case of the rich getting richer. This and other limitations might be included in mentioning some of the limitations of the project. However, that does not detract from the overall value of the research.
Author Response
Review 2
I found this to be a novel, valuable and well written application of social-construction and sensemaking theory to understanding the consequences of disasters. This is an underdeveloped area in disaster research and the authors make an important contribution.
One question I had was the extent to which a largely homogenous community may predispose the results toward a positive outcome, in contrast to not only more diverse communities, but especially communities where barriers exist between ethnic and cultural sub-groups in the community thus limiting the development of a shared narrative. This community may not have been meaningfully at risk of a more corrosive response but reflect another case of the rich getting richer. This and other limitations might be included in mentioning some of the limitations of the project. However, that does not detract from the overall value of the research.
Very good point, but there is no systematic description of the community available (besides for basic figures on employment, age, etc. which does not add any deeper knowledge about the community and its identity per se). I have therefore included this comment as a study limitation, making the reader aware of this shortcoming of the study.
Reviewer 3 Report
The paper presents a very interesting analysis of the reactions of a Community to a disaster. It is well organized, clear and full of useful references. However, the paper could be improved by an effort to make it more general, as specified in the review report.

Author Response
Review 3
The paper presents a very interesting analysis of the reactions of a Community to a disaster. It is well organized, clear and full of useful references. However, the paper could be improved by an effort to make it more general.
Given the fact that the paper addressed a specific case (a forest fire in Sweden in 2014), some points are not well addressed because they are not relevant for the case examined. But a deeper analysis could make the paper a kind of model to analyse also different cases. Particularly, the suggestion is to give more details about the following points:
Line 115 – “the cause cannot exclusively lie within nature”. The topic of the disaster cause is very important also for the reactions of the Community and could be improved;
I have included some more thoughts about this, but not in any deeper sense. To my opinion, the discussion in the next sub-section (section 2.2) makes this as a crucial issue.
Line 180 – “corrosive community”. The topic of the reactions of governmental authorities, companies, media and other extra-local actors could be extremely important in other cases than the specific one;
Agree, and this is what I try to say here; that it is a general description of disasters (and not only this particular forest fire disaster).
Line 219 – Table 1 “Therapeutic and corrosive communities”: the Table should be more general to include cases of major disasters, also when the causes are not clear or when they are man-made (like an industrial accident or a terroristic attack);
I sympathize with the comment but the table is a summary of a field and has to reflect the discussion within this field. Also, the function is to show the main differences between this two views.
Line 256 and following– The Author addressed the question of the international perspective in respect to the small wildfire compared to a mega-fire. But what about other kind of disasters?
I fully understand this broader knowledge interest (all kind of disaster, not only forest fire). What I here say – which counts for all disasters – is that a disaster should not only be evaluated in terms of biophysical consequences but more broadly. Which means that what may internationally be seen as a small disaster (in terms of size) nevertheless may be a disaster in its context. Furthermore, in this section I present the case, and to my opinion it is of most relevance to say something about wildfire disaster more generally, then begun to discuss other kind of disasters
I have also dropped this information about mega-fires (as reviewer 4 suggests).
Line 421, 422 – “The issues of accountability and processes of blame-making”. This point has to be addressed at the beginning as a general point relevant for analysis of other kinds of disasters;
Responsibilization is now mentioned in the introduction (section 1), and several times in section 2.2.
Line 440, 441 – “How a disaster is understood in terms of causes and consequences, and how organizations handle the disastrous situation, influence the affected community”. Again, this point has to be addressed at the beginning as a general point relevant for analysis of other kinds of disasters.
Done
Reviewer 4 Report
The author presents a well-written, clearly presented, and interesting case study of post-disaster dynamics (as well as the role of community narratives) concerning a recent wildfire event at a local level in Sweden. I especially appreciate the author's attention to recognizing the importance of local dynamics alongside more broad-brushed, national and regional dynamics and consequences of disaster events. The introduction of a new concept, alongside complementary and thoughtful incorporation of qualitative and quantitative data analyses in the findings and discussion, strongly position this paper for inclusion in the journal Sustainability. However, in order to strengthen this manuscript, I have offered a handful of recommendations, outlined below.
Re: lines 62-65, how is care-taking defined? This line could be sharpened a bit.
In reference to crisis (lines 79-82), how is this different from risk?
I would like to see more description of the community at the forefront of this paper. This goes back to the need to highlight preexisting conditions/dynamics at the local level before the wildfire. This does not need to be an in-depth analysis, but a recognition and explanation of how the community was characterized beforehand.
Throughout the manuscript you accurately describe that the short- and long-term dynamics following a disaster do not operate in a vacuum. However, I see no mention of preexisting dynamics and how these pre-event social, economic, historical, and political may also affect disaster outcomes for residents and other stakeholders. Even if this is not central to your manuscript's argument/purpose, it would strengthen the paper for you to recognize that these preexisting conditions also affect disaster trajectories.
Suggest removing or moving line beginning on 260, which reads "Forest fire research has introduced the concept of megafires fo wildfires that overwhelm the capabilities and endurance of available firefighting resources (Tedim et al. 2016)" as it seems out of place.
This is minor, but I would suggest renaming Section 3. "Materials and Methods" to "Research Design" given that section 3.2 is also named "Materials and Methods."
Starting on line 277, I recommend adding just a bit more detail about what the interview questions entailed--similar to what you did in lines 286-289.
Referring to Table 2. How might you explain the responses of those who felt that cohesion had not increased between neighbors?
Did the survey include validated items from previous studies? If so, please note within the text or in a footnote.
Within section 5.1, the fourth explanation could be strengthened via inclusion/referencing of previous studies that explored the effects of litigation on community cohesion/well-being. I suggest reading work by scholars Liesel Ritchie and Duane Gill on this topic, particularly with regard to the Exxon Valdez and BP Oil Spills.
Generally speaking, I think the author's arguments in the latter half of the manuscript could be bolstered with additional citations (including what I referenced above).
Re: lines 564-566, I suggest using a different phrase from "socially stronger," as that is too ambiguous.
The author nicely situates the introduction of "invented communities" within disaster literature concerning "therapeutic" and "corrosive" communities. However, the paper would be strengthened with further discussion as to how this concept differs from therapeutic community. If there are stark differences, could the author perhaps point to other events where they can speculate how invented communities may be an appropriate description? This would also bolster the usefulness of the introduced concept.
Lastly, in reference to the conclusion section of the paper, were participants asked if they believe this event made them less vulnerable or more resilient to future events? If not, it needs to be made clearer that this discussion is speculative.
I also want to recognize the author for their thoughtful description of limitations and justifications throughout the piece. Thank you for your work!
Author Response
Review 4
The author presents a well-written, clearly presented, and interesting case study of post-disaster dynamics (as well as the role of community narratives) concerning a recent wildfire event at a local level in Sweden. I especially appreciate the author's attention to recognizing the importance of local dynamics alongside more broad-brushed, national and regional dynamics and consequences of disaster events. The introduction of a new concept, alongside complementary and thoughtful incorporation of qualitative and quantitative data analyses in the findings and discussion, strongly position this paper for inclusion in the journal Sustainability. However, in order to strengthen this manuscript, I have offered a handful of recommendations, outlined below.
Re: lines 62-65, how is care-taking defined? This line could be sharpened a bit.
Revised: I think that the new examples work better to indicate how local vulnerability may be affected
In reference to crisis (lines 79-82), how is this different from risk?
The traditional distinction between crises and risk is that risk does not necessarily threaten the fundamental structure of society (in contrast to crises). I have discussed the concept risk a bit more the conceptual meaning of risk, but chosen to not have an explicit discussion on the relation between risk and crises in this paragraph (this in order to have as straight-forward flow here when carving out the research background).
I would like to see more description of the community at the forefront of this paper. This goes back to the need to highlight preexisting conditions/dynamics at the local level before the wildfire. This does not need to be an in-depth analysis, but a recognition and explanation of how the community was characterized beforehand.
Throughout the manuscript you accurately describe that the short- and long-term dynamics following a disaster do not operate in a vacuum. However, I see no mention of preexisting dynamics and how these pre-event social, economic, historical, and political may also affect disaster outcomes for residents and other stakeholders. Even if this is not central to your manuscript's argument/purpose, it would strengthen the paper for you to recognize that these preexisting conditions also affect disaster trajectories.
This is an important point, which also reviewer 2 stresses. The problem is that (as for many places hit by disasters) there is no systematic description of the community available. I have therefore instead made the reader aware of this, included this as a study limitation.
Suggest removing or moving line beginning on 260, which reads "Forest fire research has introduced the concept of megafires fo wildfires that overwhelm the capabilities and endurance of available firefighting resources (Tedim et al. 2016)" as it seems out of place.
done
This is minor, but I would suggest renaming Section 3. "Materials and Methods" to "Research Design" given that section 3.2 is also named "Materials and Methods."
done
Starting on line 277, I recommend adding just a bit more detail about what the interview questions entailed--similar to what you did in lines 286-289.
done
Referring to Table 2. How might you explain the responses of those who felt that cohesion had not increased between neighbors?
This is a good point, but it is hard from the survey to explained it, and due to that it is a rather few in relative terms (13%) I have not dug deeper into this.
Did the survey include validated items from previous studies? If so, please note within the text or in a footnote.
No we didn’t. Sweden has not experienced this kind of wildfire disaster so there is no relevant Swedish survey for this kind of disaster. We succeeded to get access to a couple of surveys from other countries, but did not use them in any systematic ways when constructing our survey (their items were not really relevant for our study).
Within section 5.1, the fourth explanation could be strengthened via inclusion/referencing of previous studies that explored the effects of litigation on community cohesion/well-being. I suggest reading work by scholars Liesel Ritchie and Duane Gill on this topic, particularly with regard to the Exxon Valdez and BP Oil Spills.
Done
Generally speaking, I think the author's arguments in the latter half of the manuscript could be bolstered with additional citations (including what I referenced above).
I hope my revision have made it stronger.
Re: lines 564-566, I suggest using a different phrase from "socially stronger," as that is too ambiguous.
I have added an explanation what is here meant by socially stronger.
The author nicely situates the introduction of "invented communities" within disaster literature concerning "therapeutic" and "corrosive" communities. However, the paper would be strengthened with further discussion as to how this concept differs from therapeutic community. If there are stark differences, could the author perhaps point to other events where they can speculate how invented communities may be an appropriate description? This would also bolster the usefulness of the introduced concept.
Added in the concluding section a connection between invented community and therapeutic-corrosive dichotomy (see section 5.2).
Lastly, in reference to the conclusion section of the paper, were participants asked if they believe this event made them less vulnerable or more resilient to future events? If not, it needs to be made clearer that this discussion is speculative.
I don’t agree here: To point of departure for this study is that social interpretations and narratives matters. The way the affected narrates the disaster, its consequences, and how the community handle this situation, influence the community identity and thereby also its robustness. According to their own experience, they have come more closely together, felt that they worked together to help each other, and this positive experience and how it is told matters. This means that I do not agree that I am speculative in the conclusion. Also, the point is not that the respondents explicit statements or beliefs in being less vulnerable, but my anlaysis of how they interpret the disaster and narrate their experience.
I also want to recognize the author for their thoughtful description of limitations and justifications throughout the piece. Thank you for your work!